# Temperature Sensitivity of Topsoil Organic Matter Decomposition Does Not Depend on Vegetation Types in Mountains

**DOI:** 10.3390/plants11202765

**Published:** 2022-10-19

**Authors:** Alexandra Komarova, Kristina Ivashchenko, Sofia Sushko, Anna Zhuravleva, Vyacheslav Vasenev, Sergey Blagodatsky

**Affiliations:** 1Institute of Physicochemical and Biological Problems in Soil Science, Russian Academy of Sciences, 142290 Pushchino, Russia; 2Agro-Technology Institute, Peoples’ Friendship University of Russia, 117198 Moscow, Russia; 3Agrophysical Research Institute, 195220 Saint Petersburg, Russia; 4Soil Geography and Landscape Group, Wageningen University, 6707 Wageningen, The Netherlands; 5Terrestrial Ecology Group, Institute of Zoology, University of Cologne, 50674 Cologne, Germany

**Keywords:** Q_10_ index, quality of soil organic matter, forest and meadow ecosystems, ungrazed and grazed land-use, mountainous soils

## Abstract

Rising air temperatures caused by global warming affects microbial decomposition rate of soil organic matter (SOM). The temperature sensitivity of SOM decomposition (Q_10_) may depend on SOM quality determined by vegetation type. In this study, we selected a long transect (3.6 km) across the five ecosystems and short transects (0.1 km) from grazed and ungrazed meadows to forests in the Northwest Caucasus to consider different patterns in Q_10_ changes at shift of the vegetation belts. It is hypothesized that Q_10_ will increase along altitudinal gradient in line with recalcitrance of SOM according to kinetics-based theory. The indicators of SOM quality (BR:C, respiration per unit of soil C; MBC:C, ratio of microbial biomass carbon to soil carbon; soil C:N ratio) were used for checking the hypothesis. It was shown that Q_10_ did not differ across vegetation types within long and short transects, regardless differences in projective cover (14–99%) and vegetation species richness (6–12 units per plot). However, Q_10_ value differed between the long and short transects by almost two times (on average 2.4 vs. 1.4). Such a difference was explained by environmental characteristics linked with terrain position (slope steepness, microclimate, and land forms). The Q_10_ changes across studied slopes were driven by BR:C for meadows (R^2^ = 0.64; negative relationship) and pH value for forests (R^2^ = 0.80; positive relationship). Thus, proxy of SOM quality explained Q_10_ variability only across mountain meadows, whereas for forests, soil acidity was the main driver of microbial activity.

## 1. Introduction

Microbial decomposition of soil organic matter (SOM) is an important source of atmospheric CO_2_ generating the climate–carbon cycle feedback [1,2]. High mountain soils are known to store a large amount of SOM, mainly caused by low mean annual temperature hampering microbial decomposition [3]. Across mountainous areas of the world, SOM stocks range widely from 31 to 310 Mg C ha^−1^ [4,5,6,7,8,9]. Importantly, most of these SOM stocks (30–65%) are concentrated in the topsoil (0–10 cm) layer [7,8] with relatively high accumulation of plant residues and slow decomposition rates [10,11]. Therefore, global warming could accelerate the decomposition rate in C-rich mountain soils and stimulate C losses as greenhouse gas CO_2_. This carbon cycle-climate feedback is exacerbated by a more pronounced increase of the average annual temperature in the mountainous and polar regions, compared to the world average [12].

The quality of SOM or its susceptibility to microbial decomposition differs between forests types due to varying contribution of forest floor and ground plant community to SOM [13] and between vegetation types, for instance, between meadow and forest [14], as well as between land use types (e.g., grazed and ungrazed meadows) [15]. The altitudinal gradient in mountains forms altitudinal zonation of vegetation. So that a vertical distribution of successive plant communities in mountains contributes to heterogeneity of SOM properties [16,17,18], which may affect temperature sensitivity of its decomposition under the climate change. The temperature sensitivity index of SOM decomposition by microorganisms (Q_10_) has been widely used to predict the soil response to climate warming [19]. This index represents changes in organic matter decomposition rate for each 10 °C of temperature increase. The magnitude of Q_10_ changes with altitude is still under debate. Some studies have shown a clear increasing trend of Q_10_ with altitude [20,21,22,23], but others have not evidenced such a trend [24,25,26]. We hypothesize that Q_10_ will change along altitudinal gradient in line with changing vegetation cover. The difference between Q_10_ values for forests and meadows will be more notable due to significant differences in quality and quantity of plant residues entering the soil. Similar shift in Q_10_ between grazed and ungrazed areas is expected because of differences in the rates of input of labile nutrients [15,27,28,29]. We selected a long transect across the five vegetation zones and short transects from grazed and ungrazed meadows to forests to consider different scenarios of Q_10_ changes. We also hypothesize that Q_10_ will increase in line with recalcitrance of SOM according to kinetics-based theory [30]. This hypothesis was tested by comparing changes in SOC quality indexes: soil C:N ratio, degree of the organic matter’s susceptibility to microbial decomposition (respiration per unit of soil carbon, BR:C) and index of microbial C assimilation calculated as ratio of microbial biomass carbon to total carbon (MBC:C) [31,32,33]. Thus, our study focuses on Q_10_ variability along the altitudinal gradient at two scales with consideration of vegetation cover and land use.

## 2. Results

### 2.1. Environmental Characteristics and Q_10_ Variation across Mountain Forests and Meadows

Along the studied mountain transects, mean air and soil temperatures did not clearly change with altitude, due to site-specific microclimate formed by the vegetation (Table 1). In subalpine meadows located at higher altitudes, the temperatures were 0.4–2.0 °C higher than under the canopy of deciduous forests. Moreover, the temperatures in deciduous forests and subalpine meadow of long transect were 1–4 °C colder than for the short ones. This fact was apparently related to the difference in altitude and topography generating local thermal circulation and leading to the microclimate differences between studied transect locations. The grass projective cover and its species richness along transects increased with altitude, i.e., from forests to meadows. The studied soils were generally C-rich with maximum values in the meadows for long and short transects. Forest and meadow soils were strongly acidic with pH values in the range 4.6–5.6. Soil C:N ratio was higher in forests than in meadows for the long transect; however, such trend was not observed for the short transects. This parameter in grazed sites was lower than in ungrazed ones; this difference can be related to additional N input to soils with livestock feces and urine. The BR:C value decreased with altitude for the long transect, but the opposite trend was found for the short transects. The MBC:C ratio ranged from 2.1 to 4.6% and had no clear patterns associated with changes in altitude or vegetation.

As expected, microbial decomposition of SOM gradually increased with rising temperature for all studied soils (Figure 1). Distinct differences between the soils under different vegetation were recorded for the decomposition rates at 22 °C, showing higher values for meadows than for forests. The Q_10_ value negligibly varied within long and short transects (Figure 2); however, this parameter differed significantly between long and short transects with average values of 2.4 and 1.4, respectively (*p* < 0.001; Welch’s *t*-test).

### 2.2. Relationships between Q_10_ Value and Environmental Characteristics

The principal component analysis (PCA) summarized the variations and relationships of the studied environmental characteristics across the investigated mountain transects. The first two PCA axes explained about 56% of the experimental data variability (Figure 3).

The first axis was associated mainly with changes in plant properties (grass cover, its richness), altitude, and soil C content, while the second axis was related to the variation of Q_10_ and BR:C values. The studied mountain sites were clearly grouped according to the scale of the transects regardless of their land use. In addition, the long transect was characterized by the substantial variability in environmental properties, with more noticeable differences between forest and meadow sites than those of the short transects.

The Q_10_ variation across all transects was positively correlated with pH and negatively correlated with BR:C (*r* = 0.45 and −0.51, Appendix A). The regression analysis taking into account vegetation, showed that BR:C and pH were the main factors influencing the Q_10_ in mountain meadows and forest, correspondingly, explaining respectively 64% and 80% of the total variance (Figure 4).

## 3. Discussion

In the studied mountains of the Northwest Caucasus, temperature increase by 10 °C will accelerate decomposition of soil organic matter by 40–150% (Figure 2). This range corresponds to observed Q_10_ values for mountainous soils globally [22,23,24,25,26]. Unexpectedly, the Q_10_ values did not differ between vegetation types regardless the spatial scale (i.e., transect length and altitudinal gradient) and land use (Figure 2). However, this parameter differed significantly between the long and short transects. So, the specificity of an individual mountain site (slope steepness, microclimate, location relative to the valley, etc.) plays a more important role in the variation of Q_10_ than the change in vegetation. Notably, drivers of the Q_10_ variability across the studied slopes differed depending on the vegetation type, i.e., BR:C for meadows and pH value for forests. The low BR:C was associated with high content of recalcitrant organic compounds (e.g., polycondensed aromatic forms) and, therefore, indicated biochemical stability of SOM to microbial decomposition [34]. Consequently, the observed negative relationship between Q_10_ and BR:C values is consistent with the Arrhenius kinetic theory, according to which decomposition of more recalcitrant organic compounds should have higher activation energies [30]. Moreover, in cold climatic conditions, soils accumulate large SOM stocks with a high relative portion of slightly decomposed plant residues (i.e., particulate organic matter fraction) consisting of macromolecular compounds (celluloses, hemicelluloses, etc.) with a high activation energy as well [10,11,35]. Therefore, across C-rich soils of mountain meadows, the highest Q_10_ was found for sites with the lowest mean annual temperature and BR:C ratio (long transect). On the contrary, across less C-rich soils of forests, pH value was the significant factor of Q_10_ variability. It largely controlled growth, activity, and structure of soil microbial communities [36,37,38]. Soil microbial biomass and its mineralization activity often increase with increasing pH value [36,38]. In addition, pH value is an important factor influencing activity of various extracellular enzymes, by changing the substrate binding and stability [39,40,41]. Therefore, it can be suggested that the pH effect on Q_10_ variation manifested indirectly via changes in the soil microbial properties.

Thus, we showed that the Q_10_ variation across mountain soils was mainly explained by the local terrain characteristics (slope steepness, microclimate, landforms) rather than by the vegetation types. For C-rich meadow soils, SOM quality (BR:C value) was the main driver of Q_10_ variability, whereas pH value controlled microbial activity in less C-rich forest soils. Understanding the dynamics of SOM in response to climate warming and its main drivers is a priority issue for timely adaptation and sustainable development of mountainous areas [42].

## 4. Materials and Methods

Study sites of mountain forests and meadows were located in the Northwest Caucasus (Russia; 43°40′–43°43′ N/40°43′–41°11′ E) and were chosen according to two scales: full vertical zonality (long 3.6 km transect) and forest–meadow ecotone (short 0.1 km transect). The long transect crossed five vegetation zones (mixed, fir and deciduous forests, subalpine and alpine meadows), and the short transects crossed two vegetation zones (deciduous forest and subalpine meadow). To consider the effect of the traditional use of subalpine meadows as pastures, we selected three short transects with grazing and the other three without grazing (ungrazed). All chosen mountain transects (one long and six short) were northeastern and had nonalkaline soil parent materials. Soils of the long transect were classified as Cambisols, Umbrisols, and Leptosols [17], and soil of short transects was Haplic (Humic) Cambisols. Along each transect, 0.5 m × 0.5 m plots were established for grass vegetation survey and soil sampling. For the long transect, three random plots were selected in each of five vegetation zones (*n* = 15). On the six short transects, the plots were established in each of two vegetation zones and, additionally, on their border—tree line (*n* = 18). At each plot, the grass projective cover and number of species (richness) were determined. After that, a composite topsoil sample (mixing five Ø 5 cm cores per plot, 0–10 cm layer: upper organo-mineral horizon) was taken. Vegetation survey and soil sampling for the long transect were carried out in August 2018, and for the short transects—in August 2020.

Soil samples were sieved through a 2 mm mesh to exclude plant roots, large debris, and stones. The samples (~50 g) were adjusted at the same moisture (60–70% of water-holding capacity) and preincubated at temperatures of 2, 12, and 22 °C for one week [43]. Then, soil subsamples were placed in vials (soil:air volume ratio of about 1:12), which were tightly closed and then incubated for 24 h at a selected range of temperatures. After that, the 1 cm^3^ air sample from each vial was collected and injected into a KrystaLLyuks-4000 M gas chromatograph (Meta-Chrom, Yoshkar-Ola, Russia) equipped with a thermal conductivity detector for measuring the CO_2_ concentration. The coefficient Q_10_ characterizing the temperature sensitivity of soil organic matter decomposition, i.e., increase of the rate of soil CO_2_ production in response to temperature rising by 10 °C, was calculated using the following equation: Q_10_ = e^10β^, where β is the slope of the equation for the exponential dependence of CO_2_ production on temperature [44,45]. The soil CO_2_ production rate at 22 °C, i.e. basal respiration (BR), was used for calculation of the BR to soil C ratio (BR:C), which reflects the SOM resistance to microbial decomposition [34]. Microbial biomass carbon (MBC) was measured by the substrate-induced respiration method [46], and then the ratio (MBC:C) was calculated. Total C and N contents in the soil samples were determined using a CHNS analyzer (Leco Corp., St. Joseph, MI, USA), then C:N ratio was calculated. The pH was measured in a soil:water suspension (1:2.5 ratio) with a conductivity meter (Sartorius Basic Meter; Göttingen, Germany).

Air and soil temperature along the studied transects were measured daily throughout the year (presented data for 2020–2021) by Thermochron iButton sensors (Maxim Integrated, San Jose, CA, USA) at 1.6 m above the ground and at 10 cm depth, respectively.

Descriptive statistics were used to determine the mean and standard error. Significant differences in the Q_10_ value across vegetation (*n* = 3 in each group) were tested using nonparametric Kruskal–Wallis ANOVA, and between long and short transects (*n* = 15 and 18) using the parametric Welch’s two-sample *t*-test. PCA was used to show variations and relationships of the studied properties, as well as to illustrate the difference between mountain sites. Prior to the analysis, the data were checked for normality by Shapiro–Wilk test and scaled to unit variance. Refinement of the relationship tightness between the properties was carried out using Pearson correlation analysis. Regression analysis was used to assess the relationship of Q_10_ with possible drivers separately for each vegetation type. Statistical analysis and visualization of experimental data were performed in R 4.1.0, R Core Team, Vienna, Austria [47].

## Figures and Tables

**Figure 1 plants-11-02765-f001:**
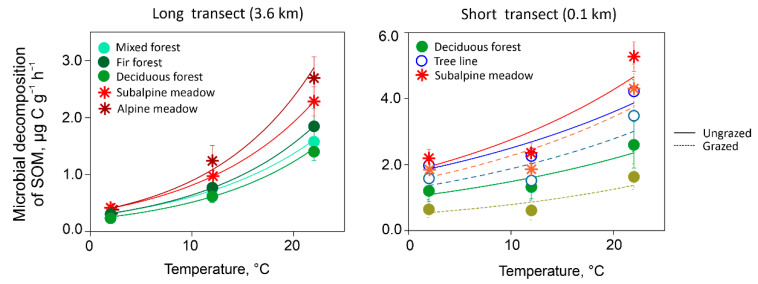
Microbial decomposition rate of soil organic matter (SOM) in relation to temperature for mountain forest and meadow sites along long (on the left) and short transects (on the right) with different land use. Symbols show means with standard error (*n* = 3).

**Figure 2 plants-11-02765-f002:**
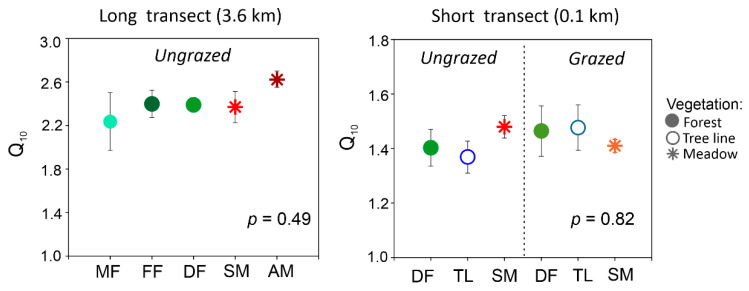
Temperature sensitivity of soil organic matter decomposition (Q_10_) of mountain forest and meadow sites along long (on the left) and short transects (on the right) with different land use. Note: MF, mixed forest; FF, fir forest; DF, deciduous forest; SM, subalpine meadow; AM, alpine meadow; TL, tree line. Symbols show means with standard error (*n* = 3). *p*-value for results of Kruskal–Wallis ANOVA.

**Figure 3 plants-11-02765-f003:**
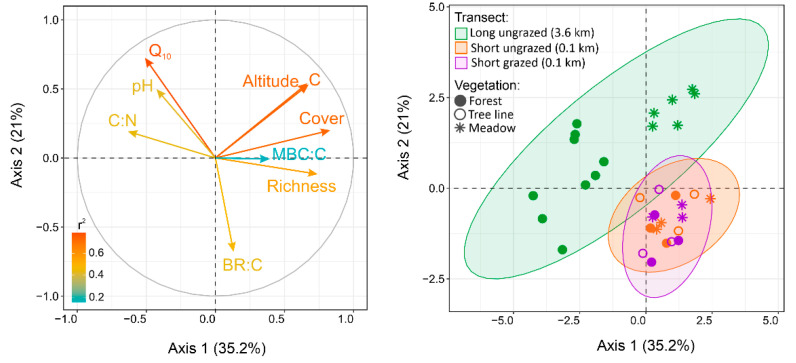
PCA results for studied plant and soil properties of mountain forest and meadow sites along long and short transects with different land use (*n* = 33). Variable correlation plot (on the left) and plot of individual sites by groups (on the right). See variable abbreviations in Table 1.

**Figure 4 plants-11-02765-f004:**
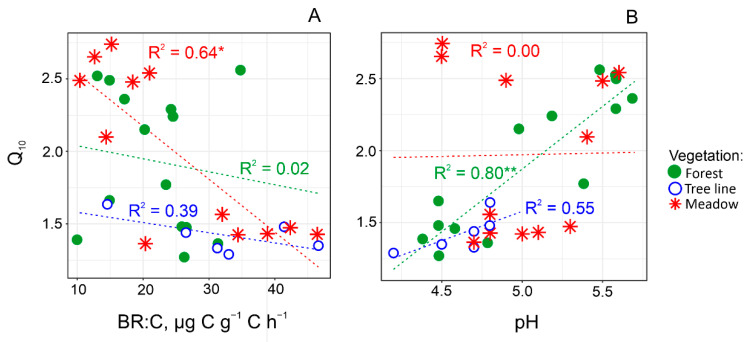
Relationship between temperature sensitivity of soil organic matter decomposition (Q_10_) and BR:C, respiration per unit of soil C (**A**) and pH value (**B**) for mountain forests and meadows along long and short transects with different land uses (*n* = 32; ** p* ≤ 0.01, *** p* ≤ 0.001. One replication for mixed forest was removed due to pH outlier).

**Table 1 plants-11-02765-t001:** Characteristics of mountain forest and meadow sites (MF, mixed forest; FF, fir forest; DF, deciduous forest; SM, subalpine meadow; AM, alpine meadow; TL, tree line) along long and short transects with different land use. Data are means ± standard error (*n* = 3).

Site	ALT, m a.s.l.	SLP, °	MAT_air_, °C	MAT_soil_, °C	Plant (Grass)		Soil (0–10 cm)				
					CVR, %	RCH	C, %	pH	C:N	BR:C, µg C g^−1^ C h^−1^	MBC:C, %
Long ungrazed transect (3.6 km)											
MF	1260	7	6.2	NA	14 ± 7	6 ± 1	5.9 ± 1.0	5.0 ± 0.4	14.7 ± 1.2	27.0 ± 2.6	2.1 ± 0.2
FF	1960	20	3.6	4.2	29 ± 11	6 ± 1	8.2 ± 1.2	5.4 ± 0.2	13.7 ± 0.6	23.7 ± 5.9	3.2 ± 0.4
DF	2060	26	4.3	3.7	18 ± 6	7 ± 0	7.9 ± 0.4	5.6 ± 0.0	13.5 ± 0.6	18.6 ± 3.3	2.7 ± 0.7
SM	2240	9	5.0	4.1	99 ± 1	8 ± 1	13.0 ± 1.6	5.5 ± 0.0	11.6 ± 0.4	17.9 ± 1.9	4.1 ± 0.2
AM	2480	6	3.6	3.7	90 ± 8	9 ± 1	21.1 ± 1.5	4.6 ± 0.1	12.4 ± 0.6	12.7 ± 1.4	2.6 ± 0.2
Short ungrazed transect (0.1 km)											
DF	2173	29	5.2	5.0	40 ± 6	8 ± 1	9.3 ± 2.4	4.6 ± 0.0	12.4 ± 1.1	26.7 ± 0.2	4.2 ± 0.7
TL	2183	27	5.1	4.8	62 ± 6	7 ± 1	11.9 ± 0.8	4.6 ± 0.2	12.3 ± 0.6	35.2 ± 3.1	3.5 ± 0.5
SM	2187	25	6.4	5.5	98 ± 2	12 ± 1	14.2 ± 0.2	5.0 ± 0.1	13.4 ± 0.2	36.2 ± 3.1	4.6 ± 1.3
Short grazed transect (0.1 km)											
DF	1884	29	7.1	6.4	53 ± 7	10 ± 0	7.7 ± 0.4	4.6 ± 0.1	11.5 ± 0.4	19.2 ± 6.5	2.9 ± 0.5
TL	1904	28	7.3	6.6	75 ± 9	9 ± 2	11.9 ± 1.5	4.7 ± 0.1	11.4 ± 0.2	29.2 ± 9.3	2.5 ± 0.1
SM	1912	29	8.5	8.4	93 ± 2	10 ± 0	11.9 ± 0.9	4.9 ± 0.1	11.5 ± 0.3	35.2 ± 7.7	2.7 ± 0.6

ALT, altitude; SLP, slope; MAT, mean annual temperature (data from 2020–2021); CVR, projective cover of ground vegetation; RCH, richness of ground vegetation (number of species per plot); C, total carbon content; C:N, ratio of total carbon to total nitrogen; BR:C, microbial decomposition rate of soil carbon; MBC:C, portion of microbial biomass carbon in total carbon; NA, not available.

## Data Availability

The data presented in this study are available within the article.

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
