# Peer review of "Temperature Sensitivity of Topsoil Organic Matter Decomposition Does Not Depend on Vegetation Types in Mountains"

_plants, 2022, doi:10.3390/plants11202765_

Round 1

Reviewer 1 Report

Abstract: Please define composition of plant community ( its species richness as well as diversity ) and quality of SOM (  what do you precisely mean , I will define in terms of proportion distribution of active versus passive pool of SOM , C: N ratio alone cannot be ultimate quality of SOM ) . How does  soil C: N ratio define the recalcitrance of  soil SOM ? .  Why do you attach so much importance to tree line transects?. R2 value of 0.04 -0.04 is extremely low , what do these values indicate at such low values ?. Eventually , what is the hometake message , if temperature sensitivity for SOM decomposition  doesn’t change alongside altitudinal  gradient ( what is the scale of this gradient  ? , is it 1220 m above msl?.) , where does  this phenomenon lead to ?. We need such conclusion at the end. The current outcome of the study doesn’t provide any novelty to our previous understanding.

Introduction : This section need reloading with literature on quality of SOM vis-a-vis plant community structure as effected by altitudinal variation.

Results : Why is there no change in soil pH ( Line 84-85) with altitudinal gradient in your case, while soil C:N ratio  along altitudinal gradient were higher in forests than in meadows ( Line 85-86). Standard deviation indicating n=3 is too low , probably that’s the reason , why no such variation is visible in your study . When is  regression considered significant (Line 105-08) , please look at R2 values., pleas elook at Figure 2 and Figure 3 ( Line 110-117) , hardly any relation exists.  

Discussion :  Any data authors generated to support such statements  (  Line 130-132 and 148-150 ) , even please specify the microbial communities involved in such microbial decomposition of SOM at various altitudes. This would have been better , had authors generated data on diversification in different microbial communities vis-à-vis temperature or altitude gradients.We need a comprehensive answer for this finding reading as : Q10 had a weak spatial variation across mountain sites 107 despite significant changes in quality of SOM ( Line 107-108).

Materials and Methods : Why did authors change the study sites an dsamplng premise during all three years of study ? ( Line 161-171). Any basis to  soil samples (~50 g) being adjusted at the same moisture (60-70% of water holding 176 capacity) (Line 176-177). Quality of SOM being defined in terms of C: N ratio is also questionable.

Conclusion : Authors have not been able to support  any regulation mechanisms of the microbial community to  temperature changes in their brief study.

Author Response

Response to reviewer 1

We appreciate the reviewer for the careful reading the manuscript and valuable comments and suggestion. We have considered all of them and corrected the text and material presentation according to reviewer’ comments. We have submitted the unmarked and marked text of manuscript by red, as well as added the supplementary material. The introduction, results, discussion and M&M were mostly rewritten. We feel hopeful that it improved the manuscript. The text proofreading (English languages corrections) was not performed. Definitely it will be done after 2-nd round of reviewing process.

Regards,

Kristina Ivashchenko

on behalf of all authors

 Comments and Suggestions for Authors

Abstract: Please define composition of plant community (its species richness as well as diversity) and quality of SOM (what do you precisely mean, I will define in terms of proportion distribution of active versus passive pool of SOM , C: N ratio alone cannot be ultimate quality of SOM ) . How does  soil C: N ratio define the recalcitrance of  soil SOM ?. Why do you attach so much importance to tree line transects?. Rvalue of 0.04 -0.04 is extremely low , what do these values indicate at such low values ?. Eventually , what is the hometake message , if temperature sensitivity for SOM decomposition doesn’t change alongside altitudinal gradient ( what is the scale of this gradient?, is it 1220 m above msl?.), where does this phenomenon lead to? We need such conclusion at the end. The current outcome of the study doesn’t provide any novelty to our previous understanding.

We have corrected the abstract considering added data, which help us find the drivers of Q10 (L 17-34).

Introduction: This section need reloading with literature on quality of SOM vis-a-vis plant community structure as effected by altitudinal variation.

We have reloaded the intro section according to the comments (L 50-55; L 62-73).

Results : Why is there no change in soil pH ( Line 84-85) with altitudinal gradient in your case, while soil C:N ratio  along altitudinal gradient were higher in forests than in meadows ( Line 85-86). Standard deviation indicating n=3 is too low , probably that’s the reason , why no such variation is visible in your study . When is  regression considered significant (Line 105-08) , please look at R2 values., pleas elook at Figure 2 and Figure 3 ( Line 110-117) , hardly any relation exists.  

The result section was reworked. It was added the proxy indexes of SOM quality according to Charro et al., 2010; Kurganova et al, 2019; Xu et al., 2014 (see reference list). Fig. 3 and Fig. 4 show results of suitable data analysis for our experimental design.

Discussion :  Any data authors generated to support such statements  (  Line 130-132 and 148-150 ) , even please specify the microbial communities involved in such microbial decomposition of SOM at various altitudes. This would have been better , had authors generated data on diversification in different microbial communities vis-à-vis temperature or altitude gradients. We need a comprehensive answer for this finding reading as : Q10 had a weak spatial variation across mountain sites 107 despite significant changes in quality of SOM ( Line 107-108).

Discussion section was re-written (L. 156-188).

Materials and Methods : Why did authors change the study sites and samplng premise during all three years of study? ( Line 161-171).

The explanation was added to introduction (L. 63-68) and some information was added to M&M for clarifying the experimental design (L. 190-197).

Any basis to  soil samples (~50 g) being adjusted at the same moisture (60-70% of water holding 176 capacity) (Line 176-177).

It was added the reference (L. 210)

Quality of SOM being defined in terms of C: N ratio is also questionable.

It was suggested additional characteristics (L. 70-72)

Conclusion : Authors have not been able to support  any regulation mechanisms of the microbial community to  temperature changes in their brief study.

The short conclusion was added to discussion section (L. 182-188).

Reviewer 2 Report

The manuscript is intresting, but quite short, and would greatly benefit from a read through for English usage and grammar.

My main concern is that the replication number is very low and the subsequent upscaling of this to an ecosystem level is questionable, please justify.

My suggestion is that this manuscript is not ready for publication without further work, additional data, etc. I am hoping that it is part of a larger project and should be put back in with those data streams.

Author Response

We appreciate the reviewer for the careful reading of the manuscript and valuable comments and suggestion. We have considered all of them and corrected the text and material presentation according to reviewer’ comments. We have submitted the unmarked and marked text of manuscript by red, as well as added the supplementary material. The introduction, results, discussion and M&M were mostly rewritten. We feel hopeful that it improved the manuscript.

 Regards,

Kristina

on behalf of all authors

Comments and Suggestions for Authors

The manuscript is intresting, but quite short, and would greatly benefit from a read through for English usage and grammar.

My main concern is that the replication number is very low and the subsequent upscaling of this to an ecosystem level is questionable, please justify.

My suggestion is that this manuscript is not ready for publication without further work, additional data, etc. I am hoping that it is part of a larger project and should be put back in with those data streams.

Thank you very much for your review. We agree that it was not enough data in previous  manuscript and the conclusion was poor. We have added the additianl data as proxy SOM quality according to Charro et al., 2010; Kurganova et al, 2019; Xu et al., 2014 (see reference list). We have decided combine the data set and it allowed us to obtain interesting results (see Fig. 3 and Fig. 4, there is presented the suitable data analysis for our experimental design). Unfortunately we hadn't opportunity the text proofreading (English languages corrections). Definitely it will be done after 2-nd round of reviewing process. We hope that this fact will not complicate the corrected manuscript evaluation.

Round 2

Reviewer 1 Report

Authors have indeed taken lot of pains ,evident from kind of rewriting they have exercised in the manuscript.  Manuscript in present form is acceptable. 

Reviewer 2 Report

N/A